# IL-18 Signaling in the Rat Central Amygdala Is Disrupted in a Comorbid Model of Post-Traumatic Stress and Alcohol Use Disorder

**DOI:** 10.3390/cells12151943

**Published:** 2023-07-27

**Authors:** Vittoria Borgonetti, Bryan Cruz, Valentina Vozella, Sophia Khom, Michael Q. Steinman, Ryan Bullard, Shannon D’Ambrosio, Christopher S. Oleata, Roman Vlkolinsky, Michal Bajo, Eric P. Zorrilla, Dean Kirson, Marisa Roberto

**Affiliations:** 1Department of Molecular Medicine, The Scripps Research Institute, La Jolla, CA 92073, USA; vborgonetti@scripps.edu (V.B.); bcruz@scripps.edu (B.C.); vvozella@scripps.edu (V.V.); sophia.khom@univie.ac.at (S.K.); mqsteinman@ucdavis.edu (M.Q.S.); rbullard@scripps.edu (R.B.); sdambros@ucsd.edu (S.D.); coleata@scripps.edu (C.S.O.); rvlkolinsky@scripps.edu (R.V.); mbajo@scripps.edu (M.B.); ezorrill@scripps.edu (E.P.Z.); dkirson@uthsc.edu (D.K.); 2Department of Pharmaceutical Sciences, University of Vienna, Josef-Holaubek-Platz 2, 1090 Vienna, Austria; 3Department of Pharmacology, Addiction Science, and Toxicology, The University of Tennessee Health Science Center, Memphis, TN 38163, USA

**Keywords:** IL-18, post-traumatic stress disorder, alcohol use disorder, CeA, sex differences, GABA

## Abstract

Alcohol use disorder (AUD) and anxiety disorders are frequently comorbid and share dysregulated neuroimmune-related pathways. Here, we used our established rat model of comorbid post-traumatic stress disorder (PTSD)/AUD to characterize the interleukin 18 (IL-18) system in the central amygdala (CeA). Male and female rats underwent novel (NOV) and familiar (FAM) shock stress, or no stress (unstressed controls; CTL) followed by voluntary alcohol drinking and PTSD-related behaviors, then all received renewed alcohol access prior to the experiments. In situ hybridization revealed that the number of CeA positive cells for *Il18* mRNA increased, while for *Il18bp* decreased in both male and female FAM stressed rats versus CTL. No changes were observed in *Il18r1* expression across groups. Ex vivo electrophysiology showed that IL-18 reduced GABAA-mediated miniature inhibitory postsynaptic currents (mIPSCs) frequencies in CTL, suggesting reduced CeA GABA release, regardless of sex. Notably, this presynaptic effect of IL-18 was lost in both NOV and FAM males, while it persisted in NOV and FAM females. IL-18 decreased mIPSC amplitude in CTL female rats, suggesting postsynaptic effects. Overall, our results suggest that stress in rats with alcohol access impacts CeA IL-18-system expression and, in sex-related fashion, IL-18′s modulatory function at GABA synapses.

## 1. Introduction

Alcohol use disorder (AUD) is a stress surfeit disorder [1], and stress is a risk factor for alcohol dependence and relapse [2,3,4]. Comorbid anxiety disorders predict more severe alcohol withdrawal symptoms [5] and increased relapse risk [6]. Animal models have helped to improve understanding of the mechanistic effects of stress on drinking [4,7,8]. Several footshock studies that model fear/anxiety [9,10] employ Pavlovian approaches in which the animal forms an association between the footshock and an otherwise neutral cue [7,11]. We recently developed an inhibitory avoidance-based shock model of PTSD that elicits increased alcohol drinking and evokes sleep disturbances, hyperarousal, fear context generalization, and irritability in rats that persist at least 9–13 weeks post-stress [12,13]. This model bolsters translational significance by using a procedure that encompasses not only Pavlovian and non-associative sensitization to fear, but also instrumental aspects related to negative reinforcement [13,14]. Models that leverage non-Pavlovian elements may demonstrate advantages such as improved construct validity when compared to those that use pure classical conditioning [15]. The model’s heightened alcohol drinking and anxiety-like phenotypes also translationally resemble psychogenic symptoms of comorbid PTSD/AUD. We reported that shock stress in either novel (NOV) or familiar (FAM) contexts increased alcohol preference in both males and females [12]. In addition, this model encompasses sex differences wherein stressed females display increased hyperarousal and decreased sleep maintenance during alcohol abstinence versus stressed males [12]. We also showed that stress and ethanol increased CeA GABAergic transmission [12]. Specifically, FAM males showed increased GABAA receptor function, while FAM females showed increased GABA release. We speculated that the heightened ethanol intake of stressed subjects may contribute to the altered CeA GABA signaling. 

Neuroinflammation is a common stress/alcohol pathway, and alcohol-induced disturbances in both pro- and anti-inflammatory pathways in the brain and blood support the “neuroimmune hypothesis” of alcohol addiction [16,17,18,19,20]. Prolonged alcohol consumption induced brain increases in proinflammatory cytokines and region-dependent increases in microglia [21,22,23,24]. Interleukin-18 (IL-18) is a member of the IL-1 family of cytokines, and it is widely expressed in brain regions involved in emotional regulation [25,26,27,28,29]. The activity of IL-18 depends on binding to IL-18 receptors, a heterodimer of the IL-18 receptor 1 ligand binding subunit (IL18R1) complex with IL-18 receptor accessory protein (IL18RAP), the subunit key for receptor signaling (formerly known as IL-18Ra and IL-18Rb subunits, respectively). Signaling is controlled by an endogenous IL-18 binding protein (IL-18BP), which inhibits excessive IL-18 activation [30,31]. An imbalance between them in the brain leads to an increase in the neuroinflammation state, which is also seen in patients suffering from generalized anxiety disorder, PTSD, and AUD [26]. There is evidence that alcohol stimulates microglial production of proinflammatory cytokines (e.g., IL-18 [24] and IL-1 β [32]) and chemokines [33,34,35,36,37]. Furthermore, a meta-analysis (human and rat) found that IL-18 system genes are the neuroimmune pathway most strongly upregulated by alcohol consumption [18,19]. Dysregulated levels of IL-18 [38,39] are also associated with major depressive episodes following stressful life events and pain [24,40,41,42,43]. Interestingly, chronic mild stress induced remarkable hippocampal microglial activation, NLRP3 inflammasome activation, and upregulation of inflammatory mediators, such as IL-1β and IL-18 [44]. Similarly, prenatal restraint stress increases the level of IL-18 in the hippocampus of male and female rat offspring, as well as levels of anxiety-related and depression-related behavior, impaired recognition memory, and diminished exploration of novel objects compared to control rats [27,29,45]. 

The amygdala is an anatomical hub that integrates stress and alcohol experiences [46,47]. The primarily GABAergic central nucleus of the amygdala (CeA) is the main output region of the amygdala complex, which plays a key role in stress and anxiety [46,48,49,50]. Chronic alcohol increases GABA transmission in the CeA across species, including rats [51,52,53,54,55,56,57], mice [58,59], macaques [60,61], and humans [62]. Furthermore, CeA GABA transmission also increased in our model of PTSD/AUD rats compared with controls [12,63]. Notably, IL-18Rα/IL-18Rβ receptor complexes are constitutively expressed on neurons in the amygdala, hypothalamus, and BNST [25,27,28,64,65,66], regulating release/activity of neuropeptides implicated in anxiety and alcohol intake [67,68]. Chronic restraint stress or footshock increases IL-18 expression [27]. Recent human studies found that (1) IL-18R gene expression is differentially associated with distinct PTSD subtypes [69], (2) a single nucleotide polymorphism (SNP) in the IL-18 gene (rs1946518) is associated with AUD in a highly traumatized civilian cohort comorbid for PTSD [70], and (3) this same SNP is also associated with increased IL-18 expression and greater amygdala reactivity in anxiety [71,72]. Thus, we hypothesized that while IL-18 is key in adaptive homeostatic regulation of normal neuronal functions in amygdala, it also contributes to the development of maladaptive stress-induced AUDs and anxiety.

Since the involvement of IL-18 in the neurobiology of PTSD and AUD remains unknown, this work aims to study IL-18 systems in the CeA of our model of PTSD and AUD comorbidity [12]. We used in situ hybridization (ISH) [53,73,74,75,76] to investigate potential alteration in the number of cells expressing gene of the IL-18 system including *Il18r1* (encodes the ligand binding subunit of the IL-18 receptor), *Il18* (encodes IL-18), and *Il18bp* (encodes for IL-18 binding protein) in the CeA of rats (both sexes) after shock stress in either novel (NOV) or familiar (FAM) contexts and alcohol drinking. Moreover, to gain insight into the functional effects of IL-18 on CeA activity, we employed ex vivo slice electrophysiology to study the effects of an acute application of IL-18 on spontaneous action-potential-independent GABAergic transmission in the CeA of these animals. 

## 2. Materials and Methods

### 2.1. Animals

We used a total of 56 adult male and female Wistar rats. A main cohort of male and female Wistar rats (total *n* = 48, 24/sex, weighing 330–400 g and 180–225 g) was purchased from Charles River (Raleigh, NC, USA). Rats (12:12 L/D cycle, food and water ad libitum) were pair-housed, separated by a perforated clear plexiglass divider to permit individual 2-bottle choice (2BC) ethanol drinking while reducing isolation stress [12]. All the 48 rats received 2BC ethanol access for 4 weeks as previously described [12]. A separate cohort of age-matched adult male Wistar rats (*n* = 8) that did not receive any 2BC ethanol access or any stress (defined as naïve) was used for electrophysiological recording to assess the effective concentration of IL-18 on inhibitory synaptic transmission in ex vivo CeA slices. All procedures followed the National Institutes of Health Guide for the Care and Use of Laboratory Animals (8th edition) and were approved by The Scripps Research Institute Institutional Animal Care and Use Committee.

### 2.2. PTSD/AUD Model Procedures 

Rats were generated using an established comorbid model of PTSD and AUD (see Figure 1a for experimental timeline). These procedures utilized an inhibitory avoidance (IA) shuttle box that was modified from a Habitest box (30.5 cm × 25 cm × 28 cm, Coulbourn, Holliston MA, USA) with a shocker as previously described [68]. The “2-hit’ shock stress was elicited using two distinguished contextual approaches. The first shock (a single 3 mA footshock administered for 2 s) was given after the rat crossed from the illuminated to the dark chamber of the IA shuttle box [12]. The second footshock occurred 48 h later, whereby half of the stressed rats received the second footstock in the same familiar apparatus as the first (termed FAM) [12]. The other half of the stressed rats received the second footshock in a novel, environmentally distinct, single-chambered apparatus (termed NOV) [12]. Controls (CTL) were handled by experimenters and were naïve to shock stress [12]. Beginning 2 weeks after the first footshock, rats received 48 h acclimation to ethanol (20% *v*/*v*) followed by chronic, intermittent (Mondays, Wednesdays, Fridays), limited 2BC access (2 h) to ethanol at scotophase onset [12]. Beginning 2 weeks after the first footshock, rats received 48 h acclimation to ethanol (20% *v*/*v*) followed by chronic, intermittent (Mondays, Wednesdays, Fridays), limited 2BC access (2 h) to ethanol at scotophase onset [12]. Two weeks later, all rats were acclimated to an initial 2-bottle choice (2BC) ethanol access (20% *v*/*v*) that lasted 48 h in their home cage. Specifically, all rats continued a 6-week regimen of 2BC intermittent ethanol access sessions. Ethanol and water consumption were calculated from bottle weights at the end of each 2 h session. Whenever ethanol was removed for nonaccess days, a second water bottle replaced it. Ethanol and water bottle positions were alternated each 2BC session [12]. FAM male rats showed significantly higher mean ethanol intake (~0.8 g/kg/2 h) (Figure 1b,c) and preference (Figure 1f,g) than CTL males (~0.4 g/kg/2 h). NOV and FAM female rats showed similar ethanol intake (~1.2–1.4 g/kg/2 h) as compared to CTL (~1.4 g/kg/2 h) (Figure 1d,e). However, FAM females showed higher preference for alcohol when compared to CTL rats (Figure 1h,i). Rats then entered a 5-week abstinence period followed by renewed post-abstinence 2BC drinking access over 3 sessions, 24 h apart, staggered across 2 weeks. The first 2 sessions were 2 h long and the final session was 24 h with drinking measured at 2 h and 24 h [12]. All rats were euthanized 24 h after the last 2BC alcohol session (therefore, at 24 h of ethanol withdrawal) for assessment of cell-type gene expression and synaptic function within the CeA. At the time of euthanasia, female rats were lavaged to determine their estrous cycle phase. 

### 2.3. RNAscope In Situ Hybridization

A subset of rats (*n* = 13; *n* = 3–4/sex/treatment, 2 CeA sections from each rat) were deeply anesthetized with 3–5% isoflurane and perfused with cold PBS/Z-fix (Anathec LTD, 6480, Battle Creek, MI, USA). Brains were rapidly removed, and immersion-fixed in Z-fix for 24 h at 4 °C, cryoprotected in sterile 30% sucrose in PBS for 24–48 h at 4 °C, flash-frozen in prechilled isopentane on dry ice, and stored at −80 °C. Brains were then sliced on a cryostat in 20 μm thick sections, mounted on SuperFrost Plus slides (Fisher Scientific, 1255015, Portsmouth, NH, USA), and stored at −80 °C. In situ hybridization was performed using RNAscope fluorescent multiplex kit (ACD, 320850) in RNase-free conditions [53,73,74,75,76]. To perform the RNAscope in situ hybridization, a target retrieval protocol was performed as suggested by the manufacturer (ACD, 320535). Slides were then submerged in target retrieval buffer at 95–98 °C for 10 min, immediately washed in distilled water and then dehydrated with 100% ethanol, and lastly digested with Protease IV for 20 min at 40 °C. Following this pretreatment, the RNAscope Fluorescent Multiplex Detection kit User Manual (ACD, doc.no. 320293) was followed and slides were mounted with DAPI-containing Vectashield (Vector Laboratories, H100, CA, USA). A negative control (ACD, 320871) was run in tandem. The probes used from ACD Biotechne were as follows: *Il18* (547731-C1), *Il18r1* (547741-C2) and *Il18bp* (547751-C3).

### 2.4. Imaging and Analysis

We took multiple (6) images from each medial subdivision of CeA at similar bregma points to control for variation using a Zeiss LSM 780 laser scanning confocal microscope (40× oil immersion, 1024 × 1024, tile scanning of CeA 5-μm z-stacks) and kept all microscope settings the same between the experiments during image acquisition. Analysis and quantification were performed as previously described [53,73,74]. Quantification was performed using Fiji, by identifying nuclei based on DAPI staining in the CeA. Nuclei were considered positive (+) for probe if corresponding fluorescent signal was present after background (negative control) subtraction. The percentage of positive nuclei for labeled cells was calculated and then normalized to CTL group. The number of *Il18bp+* nuclei that expressed *Il18+* was determined by dividing the number of co-labeled nuclei by the total of *Il18bp+* nuclei. For calculating the number of *Il18+* nuclei that co-expressed *Il18bp+*, we divided the number of co-labeled nuclei by the total of *Il18+*. Brightness/contrast and pixel dilation are the same for all representative images.

### 2.5. Ex Vivo Slice Electrophysiology

A subset of 6–8 rats per group were used for electrophysiology. Rats were anesthetized with 3–5% isoflurane and sacrificed at the beginning of the dark cycle 24 h after the last 2BC session [12]. Brains were quickly isolated and placed in ice-cold oxygenated (95% O_2_ and 5% CO_2_) high-sucrose cutting solution (in mM: sucrose, 206; KCl, 2.5; CaCl_2_, 0.5; MgCl_2_, 7; NaH_2_PO_4_, 1.2; NaHCO_3_, 26; glucose, 5; HEPES, 5) [12,53,63,74]. Coronal slices (300 µm) containing the CeA were cut on a Leica 1200S vibratome (Leica Microsystems, Buffalo Grove, IL, USA) and then incubated in artificial cerebrospinal fluid (in mM: NaCl, 130; KCl, 3.5; NaH_2_PO_4_, 1.25; MgSO_4_•7H_2_O, 1.5; CaCl_2_, 2.0; NaHCO_3_, 24; glucose, 10) at 37 °C for 30 min and then at room temperature for a minimum of 30 min. We recorded from a total of 100 neurons in the medial subdivision of the CeA, that were visualized with infrared differential interference contrast optics and CCD cameras (QImaging, Surrey, BC, Canada). Recordings were performed in gap-free acquisition mode with a sampling rate per signal of 20 kHz and low-pass filtered at 10 kHz, using a Multiclamp 700 B amplifiers, Digidata 1440 A digitizer and pClamp 10 software (Molecular Devices, Sunnyvale, CA, USA). We filled borosilicate glass patch pipettes (3–6 MΩ; Warner Instruments, Hamden, CT, USA) with internal solution (in mM: 145.0 KCl; 5.0 EGTA; 0.5 MgCl_2_; 10.0 HEPES; 2.0 Mg-ATP; 0.2 Na-GTP). Spontaneous miniature GABAA–mediated inhibitory postsynaptic currents (mIPSCs) were pharmacologically isolated with 6,7-dinitroquinoxaline-2,3-dione (DNQX, 20 µM), DL-2-amino-5-phosphonovalerate (DL AP-5, 30 µM) and CGP55845A (1 µM) and 0.5 µM tetrodotoxin (TTX). Neurons were clamped at −60 mV and experiments with a series resistance > 25 MΩ or a >20% change in series resistance, as monitored with frequent 10 mV pulses, were excluded. All mIPSC frequency, amplitude, and kinetics data were analyzed using Mini Analysis (Synaptosoft Inc., Fort Lee, NJ, USA) and visually confirmed. Only mIPSCs > 5 pA were accepted for analysis, and mIPSC characteristics were binned based on 3–5 min of recording. 

### 2.6. Statistical Analyses

Analysis of variance models (ANOVA) were used to analyze the data with stress condition (CTL versus NOV versus FAM) as a between-subjects factor. Drinking behavior was analyzed using repeated-measure ANOVA with session (1–20) as within-subjects factor and stress as between-subjects factor. Regarding RNAscope results, each probe was normalized to the naïve-stress group (CTL) and analyzed as one-way ANOVA. Electrophysiology data were analyzed with one-way ANOVA with stress condition as a between-subjects factor. In instances where ANOVA effects were significant, follow up post hoc analyses were conducted using Tukey’s HSD test, and alpha level was set at *p* < 0.05. All data graphing and statistical analysis were conducted in GraphPad Prism Version 8.

## 3. Results

### 3.1. Expression of Il18r1, Il18 and Il18bp in the CeA of Male and Female PTSD/AUD Rats

To understand how the IL-18 system in the CeA is affected by comorbid PTSD/AUD, we used ISH/RNAscope to quantify the percent of cells expressing *Il18r1* (encodes the ligand binding subunit of the IL-18 receptor), *Il18* (encodes IL-18), and *Il18bp* (encodes IL-18 binding protein) in male and female rats that experienced shock stress to a novel context (NOV) or familiar context (FAM) followed by alcohol drinking [12]. Unstressed control (CTL) drinking rats were naïve to shock stress exposure but received 2BC alcohol access [12]. We found no significant differences in CeA *Il18r1* expression across male groups (Figure 2a,b); however, although it did not reach statistical significance, we observed a slight increase in *Il18r1* expression in NOV male rats. Importantly, we observed a significant increase (Figure 2e,f; *F_2,15_ =* 7.850, *p* < 0.05) of *Il18* expression in the FAM group versus CTL and NOV groups. Notably, in male FAM groups we found a significant decrease (Figure 2c,d; *F_2,15_ =* 7.984, *p* < 0.05) in the expression of *Il18bp* as compared to the CTL and NOV rats. No significant differences were observed in *Il18* and *Il18bp* expression between CTL and NOV groups. 

Lastly, we identified an effect of stress to increase the co-localization of *Il18bp* and *Il18^+^*, whereby the number of *Il18bp*^+^ cells co-expressing *Il18^+^* was significantly higher (Figure 2g,h; *F_2,15_ =* 13.17, *p* < 0.05) in the FAM compared to the CTL and NOV rats. Interestingly, there was also a marginal difference in the FAM stress group that showed a trend toward an increase in the number of *Il18^+^* cells that co-expressed *Il18bp*^+^ (Figure 2J) as compared to CTL and NOV. This finding suggests that increased *Il18bp* co-localization may be a compensatory mechanism to counteract increased IL-18 expression.

Similarly to males, we did not observe significant differences in *Il18r1* expression (Figure 3 a,b) between CTL, NOV, and FAM groups in female rats. However, female FAM stress rats displayed a significant (Figure 3e,f; *F_2,17_ =* 7.467, *p* < 0.05) increase in *Il18^+^* cells compared to CTL and NOV female rats. Interestingly, FAM female rats showed a significant reduction (Figure 3c,d; *F_2,17_ =* 11.80, *p* < 0.05) of *Il18bp^+^* cells versus CTL and NOV groups. Moreover, the number of *Il18bp^+^* cells that also expressed *Il18* was significantly higher (Figure 3g,h; *F_2,17_ =* 17.54, *p* < 0.05) than in CTL and NOV groups. Similar co-localization was found for the number of *Il18^+^* cells that co-expressed *Il18bp*^+^ (Figure 3J) between CTL, NOV, and FAM groups. Overall, these findings support a critical interplay of IL-18BP and IL-18 in the CeA after stress in rats with a history of voluntary alcohol access, particularly in the FAM contextual environment.

### 3.2. IL-18 Reduces Vesicular GABA Release in Ex Vivo CeA Slices of Ethanol- and Stress-Naïve Male Rats

To determine the potential functional role of IL-18 on synaptic transmission, we performed whole-cell patch clamp recordings of pharmacological-isolated GABA_A_-mediated miniature inhibitory postsynaptic currents (mIPSCs) in the medial subdivision of the CeA from unstressed Wistar males with no alcohol history (naïve control). Based on previous slice physiology studies [77,78,79,80] and brain homogenates IL-18 physiological concentrations [81,82], we tested 50 and 100 ng/mL of recombinant rat IL-18 on CeA mIPSCs from naïve control. We found that 50 and 100 ng/mL IL-18 applied for 10–15 min significantly (*p* < 0.05, one sample *t*-test) decreased mIPSC frequency by 20 ± 4% and by 34 ± 4% in the CeA of naïve rats (Figure 4a,b), respectively. These data suggest that IL-18 significantly decreases vesicular GABA release. IL-18 at 50 ng/ml, but not at 100 ng/ml, also significantly decreased mIPSC amplitude (without affecting rise and decay times) by 20 ± 4% (Figure 4a,b), suggesting effects on GABA_A_ function at postsynaptic level. Overall, these data indicate that 50 ng/mL IL-18 is an effective concentration that modulates basal GABA transmission in the CeA in ethanol and stress-naïve controls. We used this concentration for the rest of the electrophysiological studies.

### 3.3. Stress Alters the Effects of IL-18 on Spontaneous Vesicular GABAergic Transmission in a Sex-Dependent Manner

We previously showed that elevated GABA_A_-receptor mediated synaptic transmission in the CeA represents a hallmark of alcohol dependence across species [46,54,61], and our 2-hit model also increased action-potential-independent mIPSCs from CeA neurons [12]. Thus, here, we used a random subset of rats (6–7) from each of three experimental groups (CTL, NOV, and FAM) for electrophysiological study (see Figure 1) to determine 1) potential baseline differences in CeA mIPSCs and 2) the synaptic effects of IL-18 on mIPSCs. Notably, CeA neurons from FAM males displayed significantly (*p* < 0.05) larger basal mIPSC amplitudes (52 ± 4 pA) compared to CTL males (44 ± 5 pA; Figure 5a,c), suggesting increased postsynaptic GABA_A_ function. Frequency (Figure 5b) and kinetics (Figure 5d,e) of mIPSCs were not different among the three groups. In females, we did not observe significant differences between groups in any mIPSC properties (Figure 5f–k), although a trend towards increased mIPSC frequency (0.95 ± 0.2 Hz; 0.74 ± 0.10 Hz; Figure 5g) was seen in NOV and FAM compared to CTL (0.61 ± 0.1 Hz) rats.

We next investigated the effects of acute IL-18 application (12–15 min) at 50 ng/ml on CeA mIPSCs of both sexes in all treatment groups (Figure 6 and Figure 7). In CeA neurons of CTL males, we found that IL-18 significantly (*p* < 0.05) decreased (84.8 ± 6.6% of baseline, *n* = 15) the frequency of mIPSCs, suggesting decreased presynaptic GABA release. Interestingly, this IL-18-induced decrease in GABA release was not observed in CeA of NOV and FAM males (Figure 6a,b). Amplitude (Figure 6c) and rise and decay times (Figure 6d,e) of mIPSCs were not altered by IL-18 application in any of the three groups. 

In CTL females, acute application of 50 ng/ml IL-18 significantly decreased (86.0 ± 4.8%; *p* < 05; *n* = 16) the frequency of mIPSCs, suggesting that it decreased presynaptic GABA release (Figure 7a,b). Notably, IL-18 also significantly (*p* < 0.05) decreased (90.2 ± 4.3% of baseline) the amplitude of mIPSCs (Figure 7a,c), suggesting that it decreased postsynaptic GABA_A_ function in CTL females. In contrast to males, IL-18 also significantly decreased CeA mIPSC frequency in NOV (to 85.9 ± 6.2% of baseline; *p* < 0.05; *n* = 14) and FAM (83.1 ± 6.6% of baseline; *p* < 0.05; *n* = 13) females. However, in both NOV and FAM females, IL-18 did not alter the amplitude of mIPSCs (Figure 7a,c) as it does in CTL, suggesting loss of postsynaptic effects in these groups. No changes were observed in the kinetics of mIPSCs (Figure 7d–e) upon IL-18 application in any of the female groups.

## 4. Discussion

There is extensive research on the effects of stress on both alcohol intake and anxiety in rodent models. Our laboratory established a two-hit model of traumatic stress to study comorbid drinking-PTSD phenotypes, elicited by past stress in familiar (FAM) or novel contexts (NOV). We previously reported that FAM and NOV stress increased alcohol preference in males and females, respectively, along with fear overgeneralization (in males) and hyperarousal and sleep disturbances (in females) [12]. This model also alters CeA inhibitory GABAergic baseline transmission in the stress-context-specific manner [12]. Here, we examined potential effects of this model of comorbid PTSD/AUD on gene expression of components of IL-18 signaling and on the functional impact of IL-18 on inhibitory CeA transmission. Our study revealed that the number of CeA expressing *ll18* was increased while *Il18bp* was decreased in FAM stressed rats of both sexes versus CTLs. FAM stressed rats also displayed higher co-expression of *Il18bp* positive cells with *Il18* as compared to CTL or NOV rats. Furthermore, exogenous IL-18 application reduced mIPSC frequencies in the CeA of CTL rats, an effect lost in both NOV and FAM males but that persisted in NOV and FAM females. IL-18 also decreased mIPSC amplitude in female CTL rats, suggesting postsynaptic effects. Our results imply that stress in rats with alcohol access differentially impacts CeA IL-18 expression and its modulatory function at the GABA synapses.

We speculated that CeA IL-18 signaling undergoes numerous changes from gene expression to proteins in our model of stress with alcohol history. This was evident by our finding that CeA *ll18* gene expression was increased after FAM stress and alcohol-drinking history. In a consistent manner, *Il18bp*, which functions to inhibit IL18 activity, was decreased in FAM stressed rats versus CTLs, suggesting a regulatory role of IL-18 signaling and highlighting a dysregulation of the IL-18 pathway in both sexes. Indeed, IL-18 acts via a receptor complex that closely resembles that of IL-1, consisting of a ligand binding protein, IL-18R1 (formerly IL-18Ra) and an accessory protein, IL-18RAP (formerly IL-18Rβ) [83]. The activity of IL-18 is controlled by endogenous soluble IL-18 binding protein (IL-18-BP), which inhibits IL18 excessive activation [83]. This action is consistent with our finding that FAM stress increases CeA *IL18* while simultaneously decreasing *IL18bp*. 

Our gene expression data revealed no significant differences in number of CeA expressing *Il18r1* across all the experimental male and female groups. We also observed that no significant changes occurred in *Il18* and *Il18bp* expression in NOV group of rats of either sex. Interestingly, we found a significant co-localization between *Il18bp* and *Il18*, in which the percentage of *Il18bp+* cells co-expressing with *Il18+* were significantly higher in the FAM stress compared to CTL or NOV group of both sexes. Note that in the FAM group the number of cells expressing *Il18bp* is significantly lower. We speculate that the increases in this *Il18bp-Il18* co-localization may represent a compensatory mechanism in response to elevated *Il18* expression. We postulate that traumatic stress and/or its resulting increases in alcohol drinking are accompanied by increases in IL-18 inflammatory action that likely leads to increases in comorbid-like behavioral phenotypes of PTSD/AUD. Future studies are needed to confirm the selective and functional role of IL-18 and receptor components in our established model of comorbid PTSD/AUD. 

IL-18 is constitutively expressed in the brain under normal and pathological conditions [24], where it influences homeostasis and behavior. We hypothesized that an imbalance in IL-18 system molecules in the brain leads to an increase in the neuroinflammatory state, which we predicted would be observed in comorbid PTSD/AUD animals. Here, at the synaptic level, we did not find significant changes in baseline action-potential-independent GABAergic transmission at presynaptic levels across both FAM and NOV stress groups compared to control in both sexes. However, a significant increase in postsynaptic GABA_A_ function was observed in FAM males, as previously reported [12]. In this cohort we did not observe increased basal vesicular GABA release in FAM compared to CTL females [12], but it is important to consider that in the present study the ethanol intake in CTL females was considerably high, and both FAM and NOV showed increased drinking, in contrast to our previous study [12]. Although voluntary intake without dependence is insufficient to alter GABAergic transmission in the CeA [84,85], we anticipated that the increased alcohol intake observed in stressed rats would contribute to the altered CeA GABA signaling. We hypothesized that the stress and alcohol history synergistically increase CeA GABA synaptic transmission, which is observed in other conditions (e.g., heightened anxiety and alcohol intake, such as alcohol dependence [46,86,87]. 

Interestingly, IL-18 application decreases vesicular GABA release in male CTL, an effect that was lost in FAM and NOV rats. In contrast, in females, the magnitude of the IL-18 presynaptic effects on vesicular GABA release is similar across CTL, FAM, and NOV groups. Furthermore, as discussed above, basal vesicular GABA release across groups was comparable, likely reflecting the comparable amount of alcohol intake between CTL and stressed females [12]. Of note, unlike the previous study [12], the current cohort of CTL females displayed high alcohol intake, reaching the drinking levels of the stressed groups. Notably, in female controls, IL-18 significantly decreased GABAA postsynaptic functions (as reflected by the IL-18 decrease in mIPSC amplitude) as it did in ethanol-naïve males in our initial study. However, this post-synaptic action of IL-18 was absent in male CTLs (who had a 2BC drinking history) and was blunted in both FAM and NOV female groups. Future studies will be conducted to understand the co-localization of IL-18 system elements with GABAergic cell types in the CeA under naïve vs. PTSD and AUD-like conditions.

A limitation of the current study is the lack of an alcohol-naïve group. We previously determined the impact of alcohol 2BC access in PTSD-like phenotypes, using a cohort of stressed rats that never had access to alcohol (they instead received 2BC access to water) [12]. We found that traumatic stress alone elicited signs of fear overgeneralization and impaired sleep maintenance, but post-stress alcohol access influenced the expression of other behaviors, exacerbating social avoidance and modifying startle and defensive behavioral responding [12]. Furthermore, at a synaptic level, FAM stress males (without history of alcohol intake) displayed significantly elevated vesicular GABA release compared to CTL males, while CeA GABA transmission did not differ between the other groups [12]. In the present study, our main goal was to study IL-18 signaling in AUD and PTSD comorbidity, and, thus, all groups were exposed to ethanol by 2BC paradigm. However, to determine the efficacy of exogenous application of IL-18 on the mIPSCs without an interference of the ethanol exposure, we used ethanol-naïve males. Indeed, application of IL-18 decreased the amplitude of mIPSCs in ethanol-naïve rats, suggesting postsynaptic effects elicited by IL-18, but not in the 2BC alcohol-exposed (CTL), rats. We speculate that the lack of IL-18-induced postsynaptic effects on GABAergic function derives from alcohol-induced changes at the GABA receptor level (e.g., receptor sensitivity and localization), which will need further investigation. 

IL-18 is a compelling target in AUD and anxiety disorders, as it modulates biological functions during stress [26] and recent studies reported greater circulating IL-18 levels in men than women during states associated with inflammation [88,89]. In addition to stress-induced expression in stress-related brain regions [27,28], IL-18 also is present in the adrenal and pituitary gland. Its expression pattern during stress suggests that peripheral IL-18 also modulates the HPA-axis [90]. IL-18 is elevated during stress through the HPA-axis, and in paracrine fashion by CRF [43], and is downregulated through the parasympathetic nervous system activation in a tissue- and mechanism-specific fashion (IL-18BP and IL-18R) [26]. Our work points to the possibility that IL-18 serves as key regulator to the sensitivity of stress- and AUD-mediated CeA GABAergic synaptic function and its potential contribution to CeA dysregulation. This work represents a first step towards understanding the alterations induced by stress and alcohol on IL-18 signaling.

Several caveats are important to contextualize the lack of change in CeA *Il18r1* ligand binding subunit mRNA expression across stress groups. First, protein levels of IL-18R1 were not assessed. Second, our study did not assess CeA Il18RAP expression, which encodes the signaling subunit of IL-18′s heterodimer receptor. Third, splice variants of both receptor IL-18R subunits with distinct actions have been identified in the CNS but were not distinguished here. For example, a shortened IL-18R1 isoform that lacks a Toll/IL-1 receptor (TIR) domain has been proposed to be an “IL-18Rα type II” decoy receptor, analogous to the type II IL-1R [25]. Similarly, a truncated splice variant of IL-18RAP encodes a soluble protein that consists of only the first immunoglobulin-like domain (EMBL/Genbank accession number AJ550893); both long and truncated forms of IL-18RAP were reported in rat brain and liver, as well as pure cultures of microglia, astrocytes, and neurons [75]. IL-18 binding to the IL-18R1 ligand binding subunit stabilizes its interaction with the signaling IL-18RAP subunit and the adaptor protein MyD88 through the TIR domain. Thus, decreases in expression of the signaling IL18RAP subunit or in relative levels of the active vs. type II decoy/inhibitory isoforms of each receptor subunit might contribute to the stress-induced blunting that we saw in IL-18′s functional effects. Finally, the level of IL-37, another endogenous ligand for the IL-18R1 receptor and IL-18BP that can influence IL-18 action [91], also was not studied. Future study of these molecular candidates may clarify the basis of adaptations in IL-18′s electrophysiologic actions. 

## 5. Conclusions

In summary, we revealed that CeA IL-18 signaling is disrupted at multiple levels in the complex IL-18 signaling cascade, particularly following traumatic stress in familiar contexts. CeA *ll18* expression was increased and the *Il18bp* component was decreased in rats with joint stress-alcohol drinking history. IL-18 normally affects CeA GABA synapses by decreasing inhibitory signaling in the CeA of stress-naïve rats, an effect that was absent in stressed males. IL-18 only decreased the postsynaptic inhibitory signaling of stress-naïve female and of stress- and alcohol-naïve male rats, suggesting that stress and/or alcohol history impairs IL-18′s postsynaptic effects. Our novel findings identify adaptations in neuroimmune IL-18 signaling within CeA GABA microcircuits in a model of PTSD/AUD comorbidity. Further work will elucidate the specific mechanisms and direct contribution of stress, drinking, or a combination of both on amygdala GABA transmission, as well as probe the IL-18 system as a candidate therapeutic target to reduce symptoms of PTSD/AUD comorbidity in both sexes.

## Figures and Tables

**Figure 1 cells-12-01943-f001:**
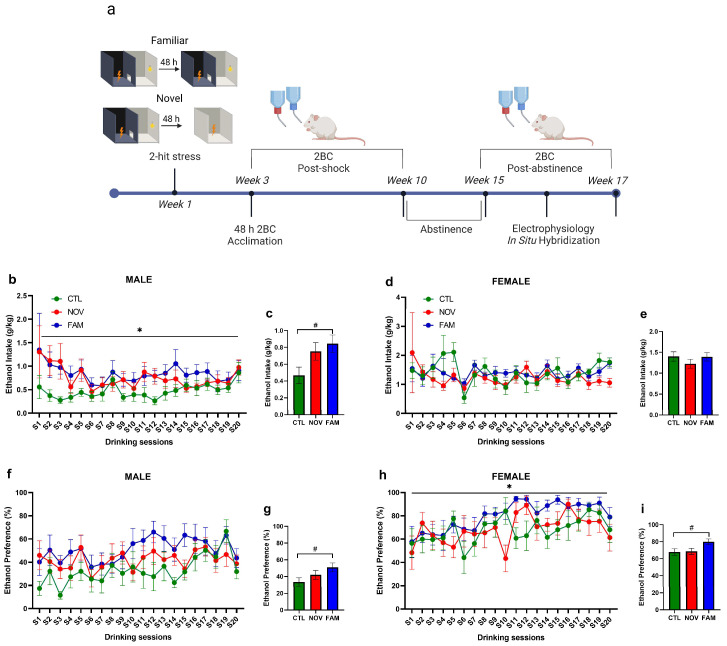
(**a**) Schematic of experimental timeline and drinking behavior. Male and female rats received either no shock (CTL) or familiar shock (FAM) or novel context shock (NOV). Two weeks later, all rats were acclimated to an initial 2-bottle choice (2BC) ethanol access (20% *v*/*v*) that lasted 48 h in their home cage. The following week all rats continued a 6-week regimen of 2BC intermittent ethanol access sessions (20%; 2 h; MWF). The rats then received a 5-week alcohol abstinence period and finally received renewed access to 2BC drinking over 3 sessions, 24 h apart, staggered across 2 weeks. The first 2 sessions were 2 h long, and the final session was 24 h with drinking measured at 2 h and 24 h later. Rats were sacrificed 24 h after their last drinking session, and tissue was processed for RNAscope in situ hybridization or ex vivo slice electrophysiology to measure IL-18 signaling in the CeA. Line graphs show daily ethanol intake (g/kg) and ethanol preference across 20 sessions in male (**b**,**f**) and female (**d**,**h**) rats. Ethanol intake and preference for each sex were analyzed by repeated measure (RM) two-way ANOVAs with stress as a between-subject factor (CTL, FAM, NOV) and daily session (S1–S20) as a within-subject factor. * *p* < 0.05, main effect of stress. Male ethanol preference (**e**) showed a marginal difference in stress, *p* = 0.079. Bar graphs reflect total mean collapsed across the 20 sessions for male (**c**,**g**) and female (**e**,**i**) intake and preference, respectively. # *p* < 0.05, pairwise comparisons between groups. Data for 16 of 1960 (0.8%) observations missing due to technical issues (e.g., bottle leakage) were estimated as the average of 10 multiple imputations (SPSS 22). All data are expressed as mean ± SEM (*n* = 8–9 rats/group). Created with Biorender.com.

**Figure 2 cells-12-01943-f002:**
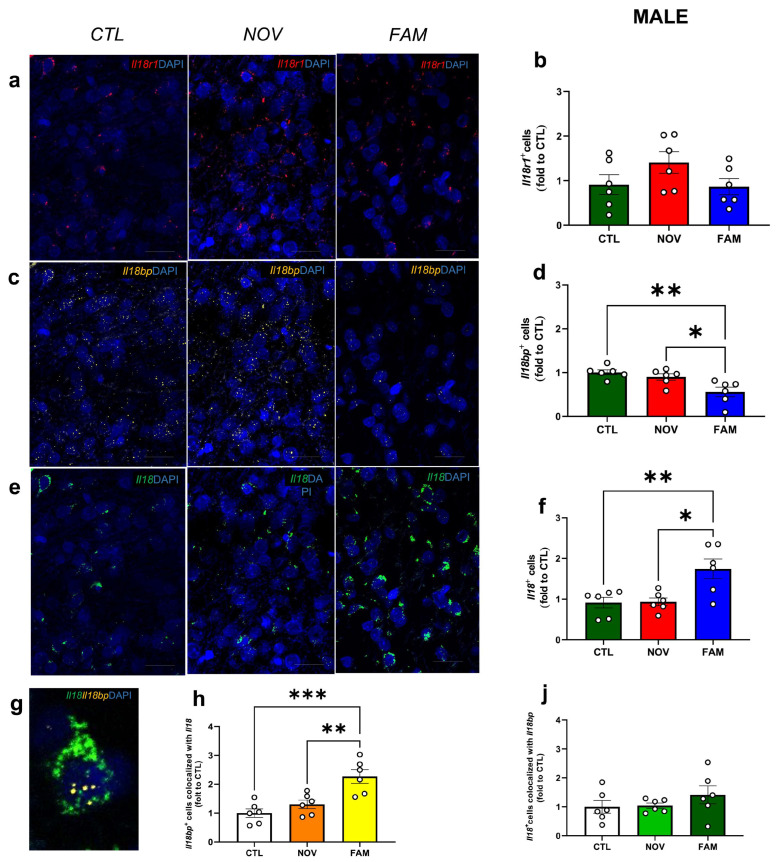
CeA gene expression of *Il18r1*, *Il18,* and *Il18bp* in males after NOV or FAM stress with alcohol history. Representative images of (**a**,**b**) *Il18r1* (red), (**c**,**d**) *Il18bp* (yellow), (**e**,**f**) *Il18* (green) and merged with the DAPI (blue) for CTL, NOV, and FAM rats. Scale bar = 10 µm. Summary graphs indicate the change in the positive nuclei expressing (**b**) *Il18r1*, (**d**) *Il18* (green), and (**f**) *Il18bp* (yellow). Representative image for (**g**) co-localization between *Il18* and *Il18bp*. Graphs reporting the analysis (**h**) for *Il18bp+* nuclei co-expressing *Il18+* and for (**j**) *Il18+* cells that co-expressed *Il18bp+.* All data are presented as mean ± SEM * *p* < 0.05, ** *p* < 0.01, *** *p* < 0.001 by Tukey’s post hoc test one-way ANOVA analysis of variance; two sections of CeA from 3 rats/group (*n* = 11–12 images). Notably, as control, we also ran representative negative control images (Appendix A).

**Figure 3 cells-12-01943-f003:**
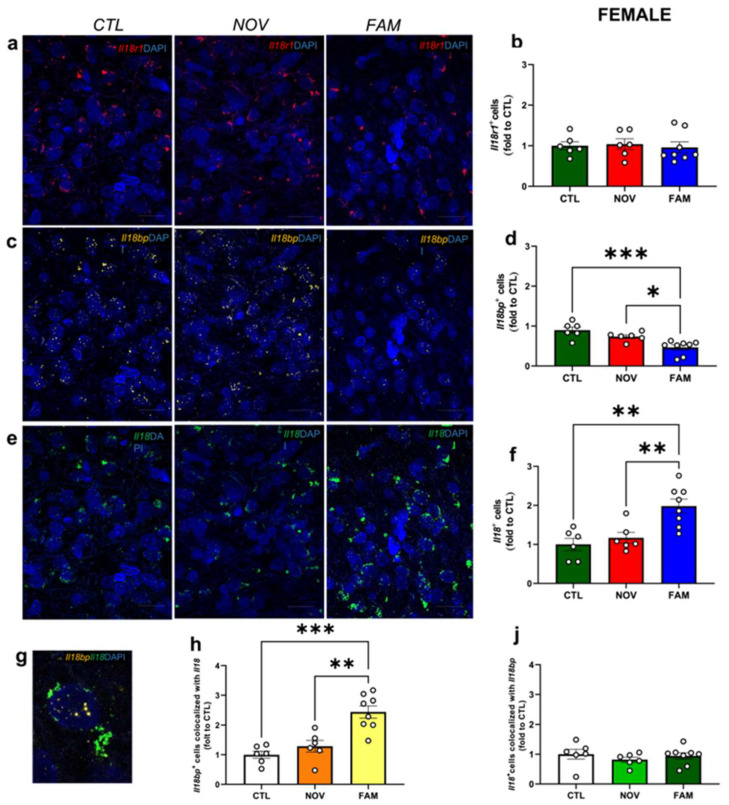
CeA gene expression of *Il18r1*, *Il18,* and *Il18bp* in females after NOV or FAM stress with alcohol history. Representative images of (**a**,**b**) *Il18r1* (red), (**c**,**d**) *Il18bp* (yellow), (**e**,**f**) *Il18* (green) and merged with the DAPI (blue) for CTL, NOV, and FAM rats. Scale bar = 10 m. Summary graphs indicate the change in the positive nuclei expressing (**b**) *Il18r1*, (**d**) *Il18bp* (yellow), and (**f**) *Il18* (green). Representative image for (**g**) co-localization between *Il18* and *Il18bp*. Graphs reporting the analysis (**h**) for *Il18bp+* nuclei co-expressing *Il18+* and for (**j**) *Il18+* nuclei that co-expressed *Il18bp+.* All data are presented as mean ± SEM. * *p* < 0.05, ** *p* < 0.01, *** *p* < 0.001 by Tukey’s post hoc test one-way ANOVA analysis of variance; two sections of CeA from 3–4 rats/group (*n* = 11–12 images). Notably, as control, we also ran representative negative control images (Appendix A).

**Figure 4 cells-12-01943-f004:**
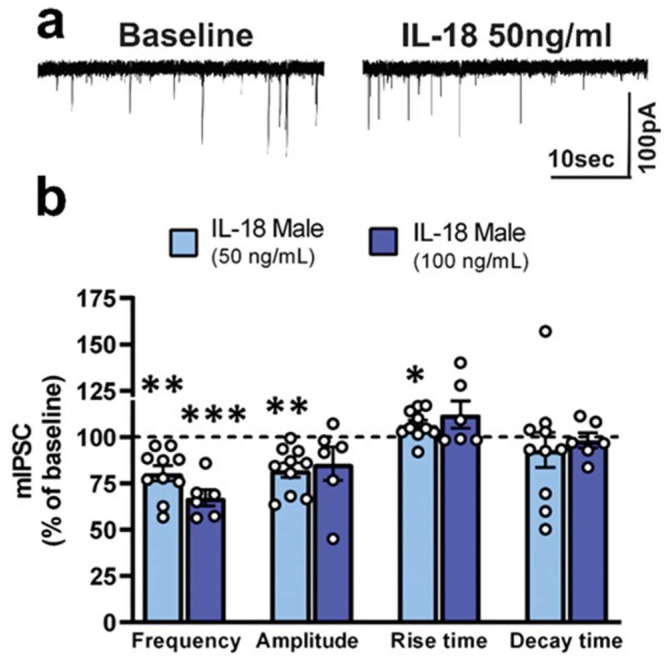
IL-18 decreases action-potential-independent GABAergic transmission in CeA of **naïve** control male rats. (**a**) Representative mIPSC before and during IL-18 application at 50 ng/mL in CeA neurons of stress- and ethanol-naïve control male rats. (**b**) Both 50 and 100 ng/mL IL-18 significantly (*p* < 0.05) decreased the frequency of mIPSCs. 50 ng/mL IL-18 also significantly (*p* < 0.05) decreased the mIPSC amplitude. Bars represent mean ± SEM; *n* = 6–10 neurons from 8 rats. * *p* < 0.05, ** *p* < 0.01, *** *p* < 0.001 by one-sample *t*-test.

**Figure 5 cells-12-01943-f005:**
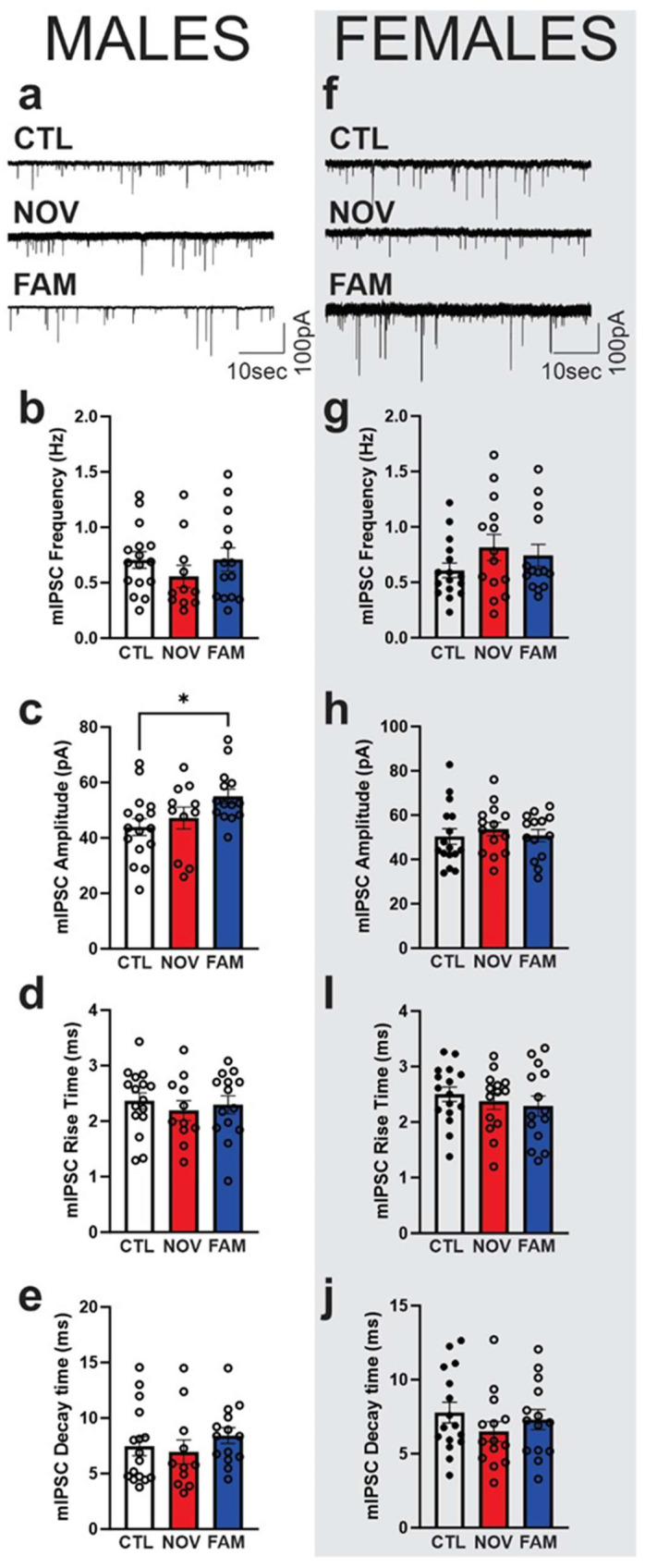
Baseline action-potential-independent GABAergic transmission in CeA all groups. Representative baseline miniature inhibitory postsynaptic currents (mIPSC) in the CeA neurons were recorded from male (**a**) and female (**f**) stress-naïve controls (CTL) and rats stressed in the novel (NOV) and familiar (FAM) environments. (**b**–**e**) All rats shared 2BC ethanol access histories as per Figure 1. Baseline mIPSC frequencies, amplitudes, and kinetics (rise and decay time) in males (*n* = 11–16 neurons). The FAM male rats showed a significant (*p* < 0.05) increase in the mean mIPSC amplitude compared to the stress-naïve control rats. (**g**–**j**) There were no significant differences in the baseline mIPSCs between the stress-naïve CTL vs. stressed female rats (*n* = 15–16 neurons). Bars represent mean ± SEM, and a significance of * *p* < 0.05 was calculated by post hoc Tukey’s multiple comparisons test.

**Figure 6 cells-12-01943-f006:**
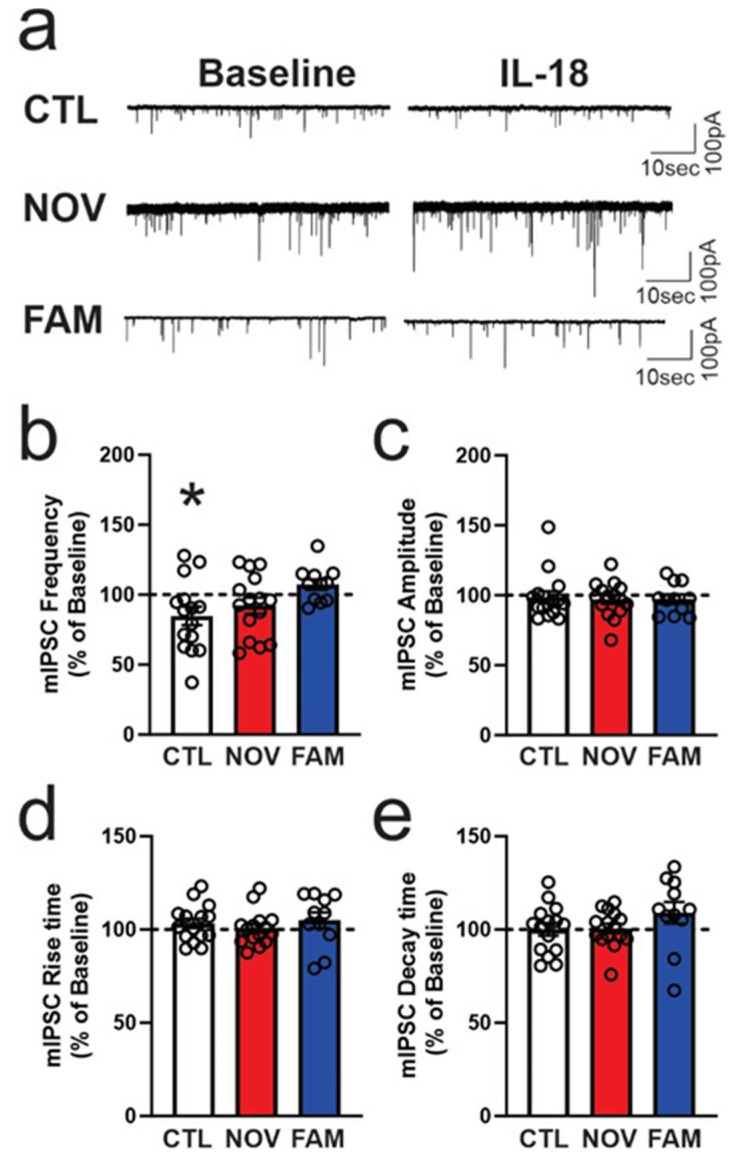
IL-18 decreases action-potential-independent GABAergic transmission in CeA of male rats. (**a**) Representative recordings of the mIPSCs from the CeA neurons before (baseline; the same traces were shown in Figure 5a) and during application of the IL-18 (50 ng/ml). The IL-18 significantly decreased the mIPSC frequencies in the stress-naïve control (*p* < 0.05; *n* = 15 neurons) but not in the stressed NOV (*n* = 15 neurons) or FAM (*n* = 11 neurons) rats (**b**). All rats shared 2BC ethanol access histories as per Figure 1. The mIPSC amplitudes and kinetics were not significantly altered by the IL-18 in the CeA neurons from all the treatment groups (**c**–**e**). Bars represent mean ± SEM, and a significance of * *p* < 0.05 was calculated by one-sample *t*-test.

**Figure 7 cells-12-01943-f007:**
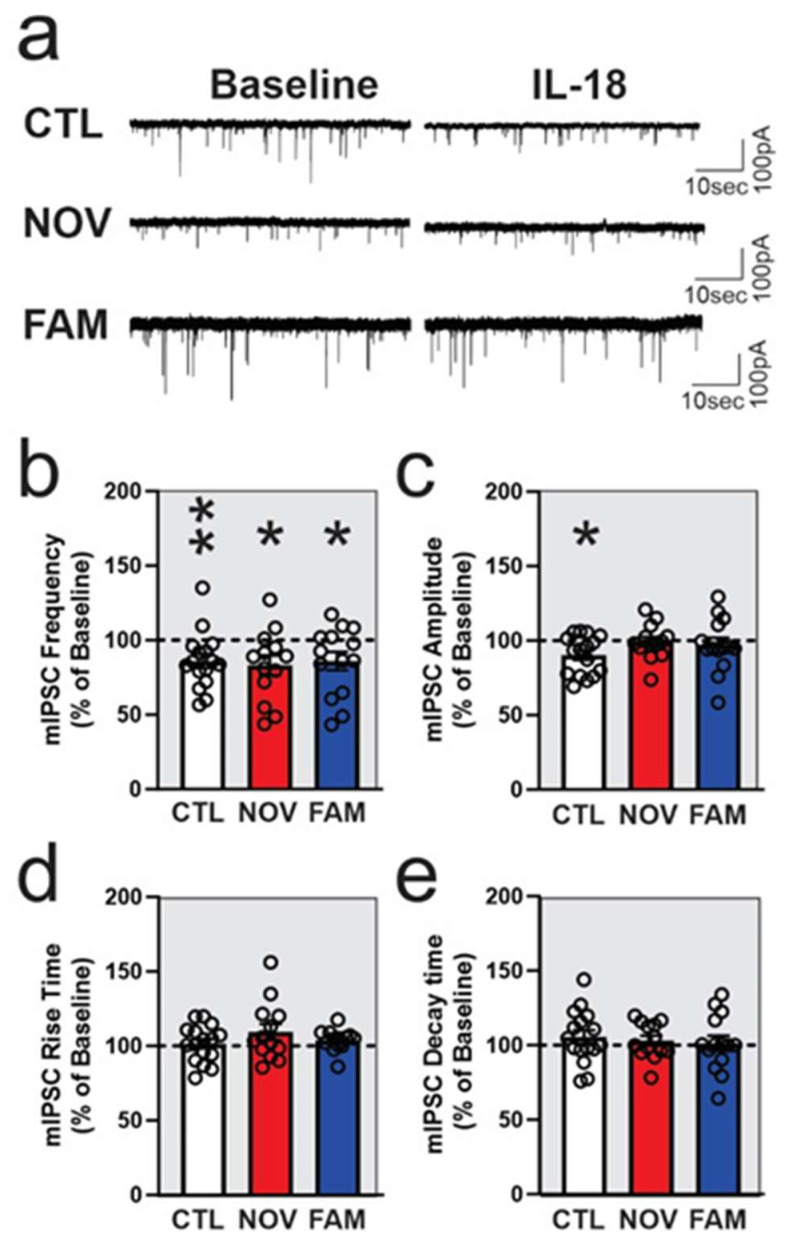
IL-18 decreases action-potential-independent GABAergic transmission in CeA of female rats. Representative mIPSCs were recorded from the CeA neurons before (baseline; the same traces were shown in Figure 5f) and during the application of the IL-18 (50 ng/mL) (**a**). The mIPSC frequencies were significantly decreased by IL-18 (50 ng/ml) in stress-naïve control (*p* < 0.01; *n* = 16 neurons) and stressed NOV (*p* < 0.05; *n* = 13) and FAM (*p* < 0.05; *n* = 11) female rats (**b**). The IL-18 significantly decreased the mIPSC amplitudes in CTL females (*p* < 0.01) but not in stressed female rats (**c**). There were no significant differences in the mIPSC kinetics elicited by IL-18 in any group (**d**,**e**). All rats shared 2BC ethanol access histories as per Figure 1. Bars represent mean ± SEM, and significances ** *p* ≤ 0.01 and * *p* < 0.05 were calculated by one-sample *t*-test.

## Data Availability

The data that support the findings of this study are available from the corresponding author upon reasonable request.

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
