# Peer review of "IL-18 Signaling in the Rat Central Amygdala Is Disrupted in a Comorbid Model of Post-Traumatic Stress and Alcohol Use Disorder"

_cells, 2023, doi:10.3390/cells12151943_

Round 1

Reviewer 1 Report

In this manuscript, Borgonetti and colleagues used a rat model of PTSD and AUD comorbidity to analyze, in the central nucleus of the amygdala (CeA) the role of the pro-inflammatory cytokine IL-18. They found, by in situ hybridization, an increase in number of cells expressing Il18 mRNA and a decrease in Il18bp in rats exposed to a familiar stress vs. unstressed control rats, and no change in Il18r1 mRNA. Using slice electrophysiology, they found that IL-18 reduced CeA GABA release, presynaptic effect that was lost in rats exposed to either a novel or a familiar stress but in males only; in addition they found that IL-18 decreased mIPSC amplitude in CTL female rats, likely a postsynaptic effect.

Overall, our results

The topic of this research is certainly timely as the role of neuroinflammation and pro-inflammatory cytokines is an important and understudied topic in both the alcohol addiction and the stress fields.  The manuscript is clearly written and straightforward and the findings are interesting.

A few minor comments that could improve the manuscript are listed down below.

- The discussion should include an interpretation of potential reasons why the CeA IL-18 signaling was disrupted particularly when the stress was applied in the familiar context. More in general, the authors should make an effort to clarify the difference (if any) between the two stress groups (FAM and NOV) in the context of PSTD-like and alcohol drinking outcomes (this information may be in the original paper describing the model, but it would be beneficial to include here for context).

- The potential contribution of the increased alcohol drinking by the two stressed groups on the ISH and electrophysiological measures should be discussed. If possible the drinking data from this cohort should be shown in a graph as well.

- In the 2.2 paragraph, specify that rats are subject to the post-shock 2BC for 6 weeks (it is in the Fig. 1 legend but not in the main text).

- Line 136: when saying “All rats were euthanized 24 hrs after the last 2BC alcohol session”, please specify if 24 hrs after the end of the session (i.e. 24 hrs of withdrawal).

- Additional discussion on the sex differences found should be included.

Author Response

In this manuscript, Borgonetti and colleagues used a rat model of PTSD and AUD comorbidity to analyze, in the central nucleus of the amygdala (CeA) the role of the pro-inflammatory cytokine IL-18. They found, by in situ hybridization, an increase in number of cells expressing Il18 mRNA and a decrease in Il18bp in rats exposed to a familiar stress vs. unstressed control rats, and no change in Il18r1 mRNA. Using slice electrophysiology, they found that IL-18 reduced CeA GABA release, presynaptic effect that was lost in rats exposed to either a novel or a familiar stress but in males only; in addition, they found that IL-18 decreased mIPSC amplitude in CTL female rats, likely a postsynaptic effect.

The topic of this research is certainly timely as the role of neuroinflammation and pro-inflammatory cytokines is an important and understudied topic in both the alcohol addiction and the stress fields.  The manuscript is clearly written and straightforward and the findings are interesting.

We thank the reviewer for her/his very positive evaluation of our manuscript and for the insightful comments that we have addressed below.

A few minor comments that could improve the manuscript are listed down below.

- The discussion should include an interpretation of potential reasons why the CeA IL-18 signaling was disrupted particularly when the stress was applied in the familiar context. More in general, the authors should make an effort to clarify the difference (if any) between the two stress groups (FAM and NOV) in the context of PSTD-like and alcohol drinking outcomes (this information may be in the original paper describing the model, but it would be beneficial to include here for context).

We thank the reviewer for giving us the opportunity to add and elaborate on the two forms of stress (FAM vs. NOV) and on the IL18 disruption. We have added information on FAM/NOV and IL-18 in the Introduction (see Page 2) as well as Discussion (see Page 17).

- The potential contribution of the increased alcohol drinking by the two stressed groups on the ISH and electrophysiological measures should be discussed. If possible the drinking data from this cohort should be shown in a graph as well.

As suggested by the reviewer we now have added the drinking data in Figure 1 and Page 4 (methods section) and we have discussed the potential contribution of drinking on the electrophysiology and RNAscope (Page 18).

- In the 2.2 paragraph, specify that rats are subject to the post-shock 2BC for 6 weeks (it is in the Fig. 1 legend but not in the main text).

We have done so.

- Line 136: when saying “All rats were euthanized 24 hrs after the last 2BC alcohol session”, please specify if 24 hrs after the end of the session (i.e. 24 hrs of withdrawal).

We have done so.

- Additional discussion on the sex differences found should be included.

As requested, we have included additional discussion on sex-differences (see Page 17-18).

Reviewer 2 Report

This manuscript reports studies of in situ hybridization and GABA transmission in central amygdala (CeA) following treatment in a model of PTSD and AUD comorbidity. The authors demonstrate that Il18 expression increased, and Il18bp decreased. Electrophysiology showed that IL-18 reduced GABA transmission, and there were sex-differences in the responses. The paper is well-written and the experiments appear to have been carefully performed. The experimental paradigm is highly novel and an interesting insight into interactions between alcohol exposure and stress exposure. Overall, it is a unique study that may have important implications for AUD/PTSD comorbidity. Enthusiasm I sreduced slightly by the following points:

 As all of the rats are exposed to alcohol, the real comparisons indicate an effect of stress on top of al alcohol-exposed baseline. The authors note that the controls had lower ethanol intakes, so it may be the lower alcohol exposure that accounts for the differences in this complex behavioral model. . The authors should explain the rationale for not including an alcohol-naïve group as a control.

 Some additional information regarding the possible role of increased drinking on the observed phenomena. Is the change in GABA transmission due to the shock paradigm, the drinking or the combination?

 More explanation/justification of the focus on IL18, in contrast to other interleukins, would be welcome. It is difficult to understand whether the authors’ observations were unique to IL18, or whether there was a change in all interleukin signaling in amygdala.

 Is it known what the concentration of IL18 is in the brain, and how the test concentrations compare to the brain concentration range?

 Minor point: “This model also alters CeA inhibitory GABAergic baseline transmission in the stress history of rats” Awkwardly worded; "stress history of rats" is unclear.

Author Response

This manuscript reports studies of in situ hybridization and GABA transmission in central amygdala (CeA) following treatment in a model of PTSD and AUD comorbidity. The authors demonstrate that Il18 expression increased, and Il18bp decreased. Electrophysiology showed that IL-18 reduced GABA transmission, and there were sex-differences in the responses. The paper is well-written and the experiments appear to have been carefully performed. The experimental paradigm is highly novel and an interesting insight into interactions between alcohol exposure and stress exposure. Overall, it is a unique study that may have important implications for AUD/PTSD comorbidity. Enthusiasm is sreduced slightly by the following points:

We thank the reviewer for her/his very positive evaluation of our manuscript and have amended our revised manuscript according to valuable suggestions. 

 As all of the rats are exposed to alcohol, the real comparisons indicate an effect of stress on top of al alcohol-exposed baseline. The authors note that the controls had lower ethanol intakes, so it may be the lower alcohol exposure that accounts for the differences in this complex behavioral model. . The authors should explain the rationale for not including an alcohol-naïve group as a control.

The reviewer raised an excellent point and we have elaborated on the lack of this group in the discussion (see Page 15). The goal of the current study was on comorbid alcohol and anxiety-like phenotype with more interest in drinking across groups. Notably, we had published data from a cohort of no-alcohol group, that paralleled those reported for 2BC animals in our Steinman et al., 2020 paper. In brief, the published results suggest that whereas many behavioral differences developed independent of ethanol (e.g., avoidance behavior in males, impaired sleep maintenance in females), some specific post-stress phenotypes may have been modified by ethanol access.  For example, the development of increased startle in females may have depended upon post-stress drinking.  In our previous studies by Steinman et al., 2020, we also reported that exposure to “2-hit” stress without history of alcohol increased CeA inhibitory GABAergic transmission only in FAM males. Notably, in that study, stress and ethanol increased CeA GABAergic transmission. Specifically, FAM males showed increased GABAA receptor function, while FAM females showed increased GABA release. Although voluntary intake without dependence is insufficient to alter GABAergic transmission in the CeA, the heightened ethanol intake of stressed subjects may contribute to the altered CeA GABA signaling. We speculated that the stress and ethanol history synergistically increase CeA GABA synaptic transmission. In the current study we do not have a control no alcohol group, thus we have acknowledged this limitation and discussed the current data in the context of our published Steinman et al., 2020 (see also next point) and see Page 17.

 Some additional information regarding the possible role of increased drinking on the observed phenomena. Is the change in GABA transmission due to the shock paradigm, the drinking or the combination?

The reviewer raised another excellent point. As discussed in the previous point, we have acknowledged that the present study lacks a no-alcohol comparison group, thus, making it difficult to tease apart the contribution of shock, drinking or a combination on GABA transmission. A statement has been added in the conclusion Page 19. A detailed discussion on GABA signaling has been added in Page 18.

 More explanation/justification of the focus on IL18, in contrast to other interleukins, would be welcome. It is difficult to understand whether the authors’ observations were unique to IL18, or whether there was a change in all interleukin signaling in amygdala.

We thank this reviewer (as well as Reviewer 1) for the opportunity to elaborate on IL-18 signaling compared to other cytokines. Please see Introduction (Page 3) and Discussion (Page 18). 

 Is it known what the concentration of IL18 is in the brain, and how the test concentrations compare to the brain concentration range?

We used IL-18 concentrations based on previous studies performed in brain slices (Francesconi, W., et al., J Neurosci, 2016. 36(18): p. 5170-80; Cumiskey, D., et al., Neuroscience Letters, 2007. 412(3): p. 206-10; Curran, B. and J.J. O'Connor, Neuroscience, 2001. 108(1): p. 83-90).

The 50 ng/ml concentration compared to the level of IL-18 measured in the brain by ELISA as in previous studies (Yatsiv, I., et al., J Cereb Blood Flow Metab, 2002. 22(8): p. 971-8; Schmidt, O.I. et al., J Neuroinflammation, 2004. 1(1): p. 13). These latter references have been added in the manuscript (Page 10)

 Minor point: “This model also alters CeA inhibitory GABAergic baseline transmission in the stress history of rats” Awkwardly worded; "stress history of rats" is unclear.

Thank you. We have corrected this sentence.